# Methamphetamine Dysregulates Macrophage Functions and Autophagy to Mediate HIV Neuropathogenesis

**DOI:** 10.3390/biomedicines10061257

**Published:** 2022-05-27

**Authors:** John M. Barbaro, Simone Sidoli, Ana Maria Cuervo, Joan W. Berman

**Affiliations:** 1Department of Pathology, Albert Einstein College of Medicine, 1300 Morris Park Ave, Bronx, NY 10461, USA; john.barbaro@einsteinmed.edu; 2Department of Biochemistry, Albert Einstein College of Medicine, 1300 Morris Park Ave, Bronx, NY 10461, USA; simone.sidoli@einsteinmed.edu; 3Department of Developmental and Molecular Biology, Albert Einstein College of Medicine, 1300 Morris Park Ave, Bronx, NY 10461, USA; ana-maria.cuervo@einsteinmed.edu; 4Department of Microbiology and Immunology, Albert Einstein College of Medicine, 1300 Morris Park Ave, Bronx, NY 10461, USA

**Keywords:** macrophage, methamphetamine, HIV-NCI, phagocytosis, ROS, autophagy, proteomics

## Abstract

HIV-neurocognitive impairment (HIV-NCI) can be a debilitating condition for people with HIV (PWH), despite the success of antiretroviral therapy (ART). Substance use disorder is often a comorbidity with HIV infection. The use of methamphetamine (meth) increases systemic inflammation and CNS damage in PWH. Meth may also increase neuropathogenesis through the functional dysregulation of cells that harbor HIV. Perivascular macrophages are long-lived reservoirs for HIV in the CNS. The impaired clearance of extracellular debris and increased release of reactive oxygen species (ROS) by HIV-infected macrophages cause neurotoxicity. Macroautophagy is a vital intracellular pathway that can regulate, in part, these deleterious processes. We found in HIV-infected primary human macrophages that meth inhibits phagocytosis of aggregated amyloid-β, increases total ROS, and dysregulates autophagic processes. Treatment with widely prescribed ART drugs had minimal effects, although there may be an improvement in phagocytosis when co-administered with meth. Pharmacologically inhibited lysosomal degradation, but not induction of autophagy, further increased ROS in response to meth. Using mass spectrometry, we identified the differentially expressed proteins in meth-treated, HIV-infected macrophages that participate in phagocytosis, mitochondrial function, redox metabolism, and autophagy. Significantly altered proteins may be novel targets for interventional strategies that restore functional homeostasis in HIV-infected macrophages to improve neurocognition in people with HIV-NCI using meth.

## 1. Introduction

Over 38 million people with HIV (PWH) are living worldwide [1]. HIV is treated with antiretroviral therapy (ART), which suppresses viral replication in infected cells but does not eradicate infection [2]. The ongoing presence of HIV in tissue reservoirs causes PWH to experience long-term comorbidities for which the underlying mechanisms are not extensively characterized [3]. One such comorbidity is HIV-neurocognitive impairment (HIV-NCI), in which PWH experience a spectrum of deficits from motor dysfunction to impaired executive function. HIV-NCI greatly diminishes quality of life and increases the overall risks of morbidity and mortality [4,5]. Approximately 15–40% of PWH have HIV-NCI, for which there are no therapies in the ART era [4,6,7].

Substance use disorder (SUD) has been shown to worsen neurocognition in people with HIV-NCI, compared to non-substance users [8,9,10]. One substance that causes long-term neurotoxicity is methamphetamine (meth), a long-acting stimulant that readily crosses the blood–brain barrier (BBB). According to the Substance Abuse and Mental Health Services Administration, over 15 million people in the United States reported meth use in their lifetimes and 2.5 million reported use in 2020—over 20% higher than in 2019. People using meth are at an increased risk of acquiring HIV [11]. They are also less likely to adhere to ART regimens [12]. A 2020 study of 351 PWH found that non-adherence to ART was ~18% higher in people who reported meth use. A recent analysis of PWH reporting meth use demonstrated higher serum levels of the inflammatory mediators, IL-6 and sD163, than in people not using meth [13]. This suggests that meth may directly impact immune cells. Thus, additional studies are needed to characterize further the interactions between meth and HIV infection in cells that mediate neurocognitive decline.

HIV enters the CNS within two weeks of peripheral infection, in part, through the transmigration across the BBB of a mature subpopulation of monocytes. These cells can differentiate into long-lived macrophages that remain in perivascular regions of the CNS for years and infect resident phagocytes, microglia, as well as other perivascular macrophages [14]. Ongoing HIV replication localizes to these cell types in post-mortem brain tissue studies of PWH who took ART long-term [15]. The infection of macrophages with HIV disorganizes the actin cytoskeleton [16]. This impairs the phagocytic uptake of extracellular proteins and apoptotic cells [17]. The higher metabolic demands of HIV-infected macrophages cause increased mitochondrial activity, which augments the production and release of reactive oxygen species (ROS) [14,18]. ROS damage healthy lipids, proteins, and DNA in other cell types, such as neurons and astrocytes [19]. HIV infection and higher amounts of ROS accelerate the rapid processing of inflammatory cytokines, such as IL-1β, that injure neurons directly [20,21].

Macroautophagy, hereby called autophagy, is a catabolic process vital to macrophage homeostasis that is also dysregulated during HIV infection [22]. Studies of people whose HIV infection advances much slower, termed elite controllers, indicate that higher levels of autophagy markers in peripheral blood mononuclear cells (PBMC) contribute to reduced disease progression [23]. Post-mortem brain tissue studies also suggest that higher levels of autophagic activity, flux, protect against HIV-mediated neurodegeneration [24,25]. In macrophages, HIV induces autophagy and inhibits the maturation of autophagosomes (APG) into autolysosomes (AL) to prevent lysosomal degradation of virions [26,27]. This causes a buildup of damaged macromolecules and dysfunctional organelles that may contribute to altered functions [28]. One key protein involved in autophagy is LC3, which is cleaved and lipidated during APG biogenesis into LC3I and LC3II isoforms [29]. LC3II present on the inner and outer membrane of the APG is degraded as maturation progresses, serving as a key marker to monitor autophagic flux [30]. Selective forms of autophagy also exist that degrade specific cargo, ranging from aggregated proteins and lipids to dysfunctional organelles, such as mitochondria and ER [31].

We hypothesized that meth enhances neurotoxicity in PWH and exacerbates HIV-NCI by further dysregulating homeostatic functions of HIV-infected macrophages. Primary human macrophages were cultured from healthy individuals, infected with HIV, treated or not with meth for 24 h, and analyzed for functional changes. We also examined the effects of a commonly prescribed ART cocktail on these functions with and without meth. Meth reduced phagocytosis of amyloid-β, increased total ROS levels regardless of baseline oxidative stress, and did not impact inflammasome-mediated IL-1β release. We found minimal changes with ART alone, and nearly all the meth-mediated changes by functional assays persisted with ART.

To determine the potential mechanisms that mediate these changes and identify the additional impacted pathways, we analyzed the proteomes of HIV-infected macrophages treated or not with meth by mass spectrometry. Meth treatment for 24 h altered the expression of proteins involved in phagocytosis and metabolism, including redox status. Given the connection of these pathways to autophagy, we also determined how meth impacted total and selective autophagic activity. Meth induced autophagy and impaired APG maturation, may reduce autophagic degradation of poly-ubiquitinated proteins, and increased autophagic clearance of mitochondria, termed mitophagy [32]. These data indicate that meth may further damage the CNS in PWH by enhancing the detrimental impact of HIV on macrophages—long-lived viral reservoirs. Differentially expressed proteins identified by mass spectrometry may be therapeutic targets to restore macrophage homeostasis and reduce neurocognitive dysfunction in PWH who use meth.

## 2. Materials and Methods

### 2.1. General Culture Methods, HIV Infection, and Key Reagents

Leukopaks were obtained from healthy donors, and PBMC were isolated using Ficoll gradient centrifugation. For proteomics, Western blotting, and RT-qPCR experiments, PBMCs were cultured directly into macrophages for 6 days in the presence of 10–20 ng/mL of M-CSF in culture media with DMEM, containing 10% FBS, 5% human serum, 1% penicillin/streptomycin, 1% glutamine, and 1% 1M HEPES on 60 mm dishes, as we described previously [33]. For phagocytosis, ROS, mitophagy, and cytokine experiments, monocytes were negatively isolated from PBMC using the MojoSort Pan Monocyte Isolation Kit (Biolegend Cat. No.: 480060, San Diego, CA, USA) according to the manufacturer’s protocols. These are commonly used methods to generate primary human MDM with a culture purity of approximately 99% by CD11b positivity [34].

After differentiation into monocyte-derived macrophages (MDM), cells were infected with 20 ng/mL of HIV_ADA_, a macrophage-tropic viral strain derived from a person with HIV (NIH, Bethesda, MD, USA). After 24 h, the excess virus was washed off and MDM were cultured for an additional 2–3 days prior to treatment with 50 µM of methamphetamine (meth) and/or ART for 24 h. This ART cocktail consists of 16 nM of tenofovir (Cat. No.: 10199), 441 nM of emtricitabine (Cat. No.: 10071), and 43 nM of dolutegravir, all obtained from the NIH AIDS Reagent Program (NIH, Bethesda, MD, USA). The concentration of meth was calculated to approximate CNS levels based on known plasma concentrations and penetrance across the BBB [35,36]. The ART concentrations tested reflect what is present in the CSF of people taking these drugs [37,38,39]. Meth, tenofovir, and emtricitabine were dissolved in water, and dolutegravir was dissolved in DMSO.

### 2.2. Measurement of HIV Infection

Supernatants from 60 mm dishes of cultured uninfected or HIV-infected macrophages after treatment with meth and/or ART for 24 h were collected and analyzed for concentrations of HIV by levels of capsid protein, p24, using a sensitive alphaLISA, according to the manufacturer’s protocols (Perkin-Elmer, Waltham, MA, USA). All samples were processed in duplicate for each experiment and averaged across all the experiments per treatment condition.

### 2.3. Mass Spectrometry Analysis

HIV-infected macrophages, either untreated or treated with meth for 24 h, were washed 3 times with cold PBS and lysed with a RIPA buffer and protein concentrations were determined. Approximately 20 µL of protein for both treatments were treated with 5 mM of DTT for 30 min and 20 mM of iodoacetamide to remove the detergents present in RIPA. Samples were then loaded in S-Trap microcentrifuge tubes and washed 3 times with a loading buffer of 90% methanol and 10 mM of NH_4_HCO_3_ and centrifuged at 500× *g* for 2 min. Pellets were moved to clean Eppendorf tubes and digested in 20 µL of 0.1 µg/mL trypsin in 50 mM of NH4HCO3 for 1 h at 47 °C. After centrifuging, the digested peptides were eluted with 40 µL of 0.2% formic acid in HPLC grade water and eluted again with 35 µL of 50% acetonitrile with 0.2% formic acid. The eluted peptides were dried for mass spectrometry analysis.

Dried samples were resuspended by a Dionex RSLC Ultimate 300 (Thermo Scientific, Waltham, MA, USA) autosampler in 10 µL of 0.1% TFA and loaded onto a 2-column setup for nano-chromatography, coupled online with an Orbitrap Fusion Lumos (Thermo Scientific, Waltham, MA, USA). Chromatographic separation was performed with a system consisting of a C-18 trap cartridge (300 µm ID, 5 mm length) and a picofrit analytical column (75 µm ID, 25 cm length), packed in-house with reversed-phase Repro-Sil Pur C18-AQ 3 µm resin. To analyze the proteome, peptides were separated using a 60 min gradient from 4–30% buffer B (buffer A: 0.1% formic acid, buffer B: 80% acetonitrile + 0.1% formic acid) at a flow rate of 300 nL/min. The mass spectrometer was set to acquire spectra in a data-dependent acquisition (DDA) mode. Briefly, the full MS scan was set to 300–1200 *m*/*z* in the orbitrap, with a resolution of 120,000 (at 200 *m*/*z*) and an AGC target of 5 × 10^5^. MS/MS was performed in the ion trap using the top speed mode (2 s), an AGC target of 1 × 10^4^ and an HCD collision energy of 35. Raw files were searched using Proteome Discoverer software (v2.4, Thermo Scientific, Waltham, MA, USA) using the SEQUEST search engine and the SwissProt human database, updated February 2020. The search included the variable modification of N-terminal acetylation, and fixed modification of carbamidomethyl cysteine. Trypsin was specified as the digestive enzyme, with up to 2 missed cleavages allowed. Mass tolerance was set to 10 pm for the precursor ions and 0.2 Da for ion products. The peptide and protein false discovery rate was set to 1%. Following the search, data were processed as previously described [40]. Briefly, the proteins were log2 transformed, normalized by the average value of each sample, and the missing values were imputed using a normal distribution 2 standard deviations lower than the mean [41].

Statistical regulation was assessed by a paired *t* test. Data distribution was assumed to be normal, but this was not formally tested. Averaging the paired *t* tests across the protein samples, we determined the differentially expressed proteins in response to meth with a fold change of ≥1.2 and ≤0.8 and significance cutoffs of *p* < 0.05 and *p* < 0.1. Using the *p* < 0.1 set of proteins, we performed KEGG pathway analysis (Kyoto, Japan) and determined the functional interactomes (String Consortium, Lausanne, Switzerland) for the cellular- and disease-related processes that meth impacted in HIV-infected macrophages. The false discovery rate (FDR) cutoffs for the KEGG pathway analysis were <0.01 and FDR < 0.05 for all the other interactome analyses. We also used a database from the laboratory of Dr. Ana Maria Cuervo (Albert Einstein College of Medicine, Bronx, NY, USA) that connected our enriched proteins to those involved in autophagy activation, regulation, and lysosomal function.

### 2.4. Phagocytosis Experiments

Negatively isolated monocytes were seeded in 24-well plates onto glass coverslips at 75,000 cells/well and differentiated into MDM for 6 days, infected with HIV, and treated or not for 24 h with meth and/or ART, as described. Then, 2–4 h prior to phagocytosis, 1 µM of Beta-Amyloid (1–42), HiLyte™ Fluor 488 peptide was aggregated at 37 °C in phenol-red free media (Anaspec Cat# AS-60479-01, Fremont, CA, USA). Amyloid solution was added to untreated (Untx) or treated MDM for 2 h. Coverslips were washed 2–3× with PBS, fixed with 3.7% PFA, washed 3 times with PBS, and mounted on frosted microscope slides with ProLong Diamond Antifade Mountant with DAPI (Invitrogen Cat# P36962, Carlsbad, CA, USA). The slides were dried at room temperature for at least 24 h and then visualized on a Zeiss CLEM by an observer blinded to the treatment condition. Fifty to seventy-five cells per treatment were analyzed in ImageJ for the intensity of green, fluorescent signals per cell. Median fluorescence per treatment in each experiment was averaged across all experiments, converted to a fold change in amyloid accumulation relative to the untreated control (HIV Untx) set to 1.0, and tested for statistical significance. A set of control cells were treated with DMSO or pre-treated with 20 µM of actin polymerization inhibitor, cytochalasin D (CytD), for 20 min with CytD present during amyloid phagocytosis (Tocris Bioscience Cat#1233, Minneapolis, MN, USA) [42]. After fixation and washing, DMSO and CytD-treated MDM were permeabilized for 5 min with 0.1% Triton-×-100 in PBS and incubated for 30 min with Alexa 594-conjugated phalloidin to stain the cytoskeleton (Invitrogen Cat#A12381, Carlsbad, CA, USA). The coverslips were washed 2 times with PBS, mounted, and visualized as described.

### 2.5. Reactive Oxygen Species Measurement

Monocytes that were negatively selected were cultured on 96-well plates (Corning Cat#3603, Corning, NY, USA) at 100,000 cells per well for 6 days to differentiate into MDM, infected with HIV, and treated with meth and/or ART for 24 h, as described. MDM were then stained for 1 h with 10 µM of CM-H2DCFDA (Invitrogen Cat#C6287, Carlsbad, CA, USA) that fluoresces upon coupling to all the reactive oxygen species (ROS) present in the cell. Post-staining, the cells were washed with PBS or incubated with 30 µM of CCCP (Sigma-Aldrich Cat#C2759, St. Louis, MO, USA), a mitochondrial uncoupler, or 5 mM pf N-acetyl-L-cysteine (NAC), an ROS neutralizer (Sigma-Aldrich Cat#A725, St. Louis, MO, USA) [43]. The fluorescent signal from the reacted DCF was read with excitation at 495 nm and emission at 520 nm by a SpectraMax M5 microplate fluorometer and analyzed in SoftMax Pro (Molecular Devices, San Jose, CA, USA). In each experiment, the fluorescent signals from sextuplet wells per treatment were averaged and compared to the untreated control, which was set at 1.0 in each experiment.

For the autophagy manipulation experiments related to ROS, in the last 4 h of treatment, some untreated or meth-treated cells were incubated with 200 µM of leupeptin, which inhibits lysosomal degradation of autophagic cargo (Fisher Scientific Cat#BP2662) [44]. Other untreated or meth-treated MDM were incubated with 500 nM of Torin 2, an inhibitor of mTORC1 that leads to classical autophagy induction, in the last 3 h of treatment (Tocris Bioscience Cat#42-481-0, Minneapolis, MN, USA) [44]. This timing and concentration of Torin 2 was optimized to ensure the induction of autophagy in HIV-infected MDM, increased LC3II flux, and decreased p62 levels. Fluorescence from DCF was read, normalized, and quantified as described.

### 2.6. Autophagic Flux and p62 Experiments

HIV-infected MDM were exposed to meth and/or ART or not for 24 h. To assess LC3II and p62 flux, lysosomal inhibitors, 20 mM of NH_4_Cl + 200 μM of leupeptin (N + L), were added to some cells in the last 4 h of exposure. Concentrations of these inhibitors were optimized as described previously [33]. After treatment, the dishes were washed with PBS at 4 °C and cells were lysed with RIPA buffer and 1:100 Halt Protease Inhibitor and Phosphatase Inhibitor (Thermo Fisher Scientific Cat#78425, Waltham, MA, USA). For RT-qPCR, RNA was isolated by Trizol, according to the manufacturer’s protocols (Thermo Fisher Scientific, Waltham, MA, USA). Lysates were analyzed for protein concentration using Protein Assay Reagent Concentrate (Bio-Rad, Hercules, CA, USA). A total of 35–50 μg protein/lane supplemented with β-mercaptoethanol was resolved by SDS-PAGE, and the protein was transferred onto nitrocellulose membranes overnight at 4 °C (GE Healthcare, Chicago, IL, USA). Revert Total Protein Stain (Li-Cor, Lincoln, NE, USA) was added to the membranes according to the manufacturer’s protocols to measure the total protein levels using an Odyssey Fc machine (Li-Cor). The signal was visualized in the dynamic range. Total protein was confirmed as a reliable Western blotting loading control in our system of primary human macrophages, as described previously. The optical density of all the signals was quantified in Image Studio v.5.2 software (Li-Cor). The membranes were then blocked in nonfat dry milk 5% in TBS with 0.1% Tween-20, followed by incubation overnight at 4 °C with rabbit antibodies to LC3B (52 ng/mL, 1:1000 Cell Signaling Technology #2775, Danvers, MA, USA) or p62 (0.5 μg/mL, 1:1000 Enzo Life Sciences BML-PW9860, Ann Arbor, MI, USA). HRP-fixed goat anti-rabbit antibody (65.7 ng/mL, 1:1000 Cell Signaling Technology #7074) was then added for 1 h. The signal was developed using Super SignalWest Femto Chemiluminescent Substrate and Luminol/Enhancer according to the manufacturer’s protocol (Thermo Fisher Scientific, Waltham, MA, USA), quantified, and normalized to total protein.

A total of 2 µg of RNA for each treatment per experiment was reverse transcribed into cDNA by SuperScript Vilo Master Mix (Invitrogen, Carlsbad, CA, USA) and stored at −20 °C. Taqman Gene Expression Assays (Applied Biosystems, Waltham, MA, USA) for *18S* reference (Cat#Hs99999901_s1, Thermo Fisher Scientific) or *SQSTM1* (Cat#Hs01061917_g1, Thermo Fisher Scientific, Waltham, MA, USA) were performed in the Taqman Gene Expression Master Mix on a StepOne Plus Real-Time PCR system (Applied Biosystems, Waltham, MA, USA). The relative quantity of *SQSTM1* in infected MDM treated with meth (HIV Meth) was measured by 2^−ΔΔCt^ relative to the control (HIV Untx), with *18S* as reference. The fold change in cDNA quantity per treatment was calculated relative to the control set to 1.0 in each experiment.

To determine the detailed changes in autophagy in response to meth and ART, LC3II or p62 was analyzed in the following three ways: steady state protein levels, rate of autophagic activity, flux, and amount of autophagic activity, net flux [33]. These parameters were calculated according to previously used methods. Lower steady-state LC3II and low flux would indicate dampened autophagy induction. High steady-state LC3II and high flux suggest induced autophagy, met with appropriately increased maturation. High steady-state LC3II with lower flux would suggest only reduced maturation, and high steady-state LC3II with minimal change in flux would indicate induced autophagy not matched with higher maturation.

### 2.7. LC3 and TOM20 Immunofluorescence Studies of Mitophagy

HIV-infected MDM were cultured at 75–100,000 cells per well onto glass coverslips in 24-well plates. MDM were treated with meth or not for 24 h, and N + L at the mentioned concentrations were added in the last 4 h of treatment to calculate the mitophagic flux. After treatment, the cells were washed 3× with PBS, fixed with 3.7% PFA, washed, and permeabilized for 2 min with 0.1% Triton-×-100 in PBS. Coverslips were incubated for at least 30 min with a blocking solution and exposed to primary antibodies diluted in blocking solution overnight at 4 °C for LC3B (520 ng/mL, 1:100 Cell Signaling Technology #2775, Danvers, MA, USA) and TOM20 (2 μg/mL, 1:100 Santa Cruz Biotechnology F-10 sc-17764, Dallas, TX, USA) or mouse IgG2a (25 μg/mL, 1:500 Invitrogen Cat# 02-6200, Carlsbad, CA, USA) and rabbit IgG (2 μg/mL, 1:250 Invitrogen Cat# 02-6102, Carlsbad, CA, USA). Coverslips were washed and incubated for 1 h at room temperature in antibodies diluted in the same blocking solution, goat anti-rabbit Alexa Fluor 488 (4 μg/mL, 1:500 Invitrogen #A-11008, Carlsbad, CA, USA) and/or goat anti-mouse Alexa Fluor 594 (4 μg/mL, 1:500 Invitrogen #A-11005, Carlsbad, CA, USA). Cells were washed with PBS and mounted onto frosted microscope slides with ProLong Diamond Antifade Mountant with DAPI and dried for at least 24 h at room temperature (Invitrogen, Carlsbad, CA, USA).

The cells were visualized at 40× in Z-series by confocal microscopy (Leica DMI8) by a blinded observer. The images were analyzed in Volocity 64 (Quorum Technologies, Lewes, UK). The number of LC3 puncta, total mitochondrial volume, and colocalization of LC3 with TOM20 were measured in each Z-series. The total red signal and number of puncta were quantified in 40–80 cells per treatment with thresholds for puncta size and intensity, as well as TOM20 signal, determined individually per experiment. LC3 puncta positive for mitochondria, which appear yellow upon overlay, represented the mitophagosomes that were also quantified in response to each treatment. LC3 flux and mitophagic flux were calculated as in Western blotting experiments. The number of puncta in cells with N + L was divided by the number of puncta in cells without N + L per treatment, and this was set to 1.0 for the untreated condition in each experiment. The fold change in mitophagic flux upon meth treatment was then calculated in each experiment and averaged.

### 2.8. Statistical Analyses for Functional Assays

All the quantitative data were analyzed in Microsoft Excel or Prism software v.9.0.1 (GraphPad Software Inc., San Diego, CA, USA). Shapiro–Wilk tests with *p* = 0.05 as the cutoff were used to test the data for normality. All data were normally distributed when more than two groups were compared. For all the two-group analyses, appropriate unpaired or paired Student’s *t* tests were used. When more than two groups were analyzed for significance, one-way ANOVA was used. When ANOVA was performed, this was followed by a multiple-comparison Dunnett’s test or a Turkey’s test to determine the differences between specific groups. One-sample *t* tests and Wilcoxon signed rank tests were used to test the fold changes for significance depending on normality. The untreated (Untx) control was set to 1.0 for each experiment. Values of *p* < 0.05 were considered statistically significant for all functional assay experiments.

## 3. Results

### 3.1. Methamphetamine Alters Protein Expression in HIV-Infected Macrophages

Monocyte-derived perivascular macrophages are resilient reservoirs for HIV in the CNS that perpetuate ongoing neuroinflammation and CNS damage [14]. We tested whether methamphetamine (meth) changes the functions of HIV-infected macrophages that mediate CNS dysfunction. Mass spectrometry was then performed to identify the proteins that meth changes that may mediate functional differences. We cultured primary human macrophages from four individual donors, infected with HIV for 24 h, washed off virus and cultured for an additional 2–3 days, and left them untreated (HIV Untx) or treated with 50 µM of methamphetamine (HIV Meth) for 24 h. Lysates were collected and subjected to mass spectrometry analysis to examine the global changes at the protein level, which were mapped to both the human and HIV proteomes. Raw protein data per experiment were normalized with negligible changes in spread, indicating high quality (Figure 1A,B). Each read was analyzed for statistical significance using paired *t* tests with identification of 90 differentially expressed proteins, using a *p*-value cutoff of 0.05, log2(4.32), and fold changes of ≥1.2, log2(0.26), or ≤0.8, log2(−0.32). Of these, 43 proteins were significantly upregulated and 47 were significantly downregulated (Figure 1C).

An important internal control for our proteomic analyses was the detection of HIV viral proteins. Mass spectrometry detected five total reads representing HIV proteins, Gag, Gag-Pol, or gp160, which were found in all four experiments. The levels of these reads were plotted as ranked abundance per protein relative to the 3655 total reads detected in each individual experiment. This enabled the visualization of the variability in protein levels across the primary human samples. We also used a p24 alphaLISA assay on the culture supernatants from these same four experiments to detect levels of HIV in pg/mL. By alphaLISA, there was a trend toward increased HIV levels with meth treatment, although this was not significant statistically. Overall HIV infection was highest in MDM from the experiments, represented by the green and purple dots (Appendix A). Importantly, HIV protein reads were highest in the untreated and meth-treated cells from the same two experiments and lowest in those represented by the black and brown dots. The trends in HIV infection with meth treatment in each experiment also corresponded with the changes in each read detected by mass spectrometry (Appendix A). Although there were upward trends for HIV proteins with meth treatment, these were not significant statistically. These data indicate that our proteomic analyses are consistent with an independently performed experiment.

### 3.2. Meth Inhibits Phagocytosis and Alters Expression of Phagocytic Proteins

Phagocytic clearance by macrophages of neurotoxic proteins and apoptotic cells is essential for maintaining a stable CNS environment [45]. Increased levels of aggregated amyloid-β in the CNS have been associated with neurodegeneration and HIV neuropathogenesis [46,47]. Monocyte-derived macrophages (MDM) were cultured from PBMC on glass coverslips, infected with HIV as described, and left untreated (HIV Untx) or treated with 50 µM of meth and/or ART (16 nM of tenofovir, 441 nM of emtricitabine and 43 nM of dolutegravir) for 24 h. We measured the phagocytosis of aggregated, fluorescent 1–42 amyloid-β peptide in untreated HIV-infected MDM and in response to meth and/or ART. An amount of 20 µM of cytochalasin D (CytD), an inhibitor of actin polymerization essential for phagocytic engulfment, was a control (Figure 2A) [42]. Meth alone significantly decreased phagocytosis of aggregated amyloid-β peptide approximately 30% relative to the control. ART alone had no significant effect, yet interestingly, meth + ART appeared to increase phagocytosis relative to meth alone (*p* = 0.054, Figure 2B). These ART drugs may oppose the mechanisms by which meth inhibits phagocytosis. Reduction in phagocytosis is not likely to be due to drug toxicity, since meth treatment of up to 1 mM has been shown to be non-toxic to primary human monocytes and macrophages [35,38,48].

To determine the potential mechanisms of phagocytic suppression by meth, we identified the differentially expressed proteins related to the actin cytoskeletal arrangements important for substrate engulfment. Significant decreases in three factors that promote actin organization, *ARHGEF9*, *PALLD*, and *ASAP1*, were detected [49,50]. Meth also increased levels of *RCC2*, which reduces arrangement [51]. There was also a trend toward decreased *ELMO1*, which is required for actin organization during phagocytic engulfment (Figure 2C) [52]. We found an increase in *FCGR3A* and trends toward decreased *MSR1* and *SCARB*, the receptors involved in recognizing cargo to internalize (Figure 2C) [53]. Meth may both disorganize the actin rearrangements necessary for substrate engulfment and impact the recognition of substrates at the cell surface.

Phagocytosis is a dynamic, multi-step process regulated by signaling. Meth significantly increased calcium signaling effectors, calmodulin (*CALM1/2/3*) and *CAM2KD*, which activate the proteins that contribute to cell motility and contraction, such as *PRKCA* (Figure 2D) [54,55]. Membrane lipids enriched for PI(3)P anchor proteins that promote trafficking of phagocytic vesicles [50]. Meth significantly decreased *PIK3C3*, Vps34, a vital kinase that produces PI(3)P lipids [56]. There was also a trend toward decreased levels of *PIK3R1*, which positively regulates PI3K activity (Figure 2D) [56]. This suggests that meth may inhibit phagocytosis by altering the signaling necessary for phagocytic function.

### 3.3. Meth Increases ROS and Changes Expression of Proteins Involved in Related Metabolic Processes

We also measured the effects of meth on reactive oxygen species (ROS) levels because released ROS are toxic to other CNS cells. MDM were cultured on 96-well plates, infected with HIV, and treated or not with meth and/or ART for 24 h. Total cellular ROS were stained using a DCF-based fluorescent dye. Either PBS or 30 µM of CCCP in PBS was added 1 h prior to quantifying the fluorescence in sextuplet wells per condition in each experiment. CCCP was added to determine whether an effect on ROS by meth and/or ART could persist with already increased mitochondrial stress, which exists in the CNS of people with HIV-NCI [6]. N-acetylcysteine (NAC), an ROS neutralizer, was the additional control (Figure 3A). Meth with or without ART significantly increased ROS with both PBS alone and CCCP as the baselines (Figure 3A,B). The increase with meth + ART was significantly higher than with ART alone but not higher than meth alone, suggesting that the effect of meth is predominant.

By proteomics, we detected significant increases in electron transport chain (ETC) factors, *COX7A2* and *UQCRSF*, and a decrease in ETC components, *COX6A1* and *NDUFB7* (Figure 3C) [57]. While not statistically significant, there were also trends toward increased citric acid cycle-related enzymes, *IDH1*, *IDH2*, and *PDHB* (Figure 3C) [58]. A significant increase in *L2HGDH* may also increase the levels of α-ketoglutarate, a metabolite also produced by isocitrate dehydrogenase (Figure 3D) [59]. This may reflect the changes in aerobic metabolism that mediate increased ROS production, as well as mitochondrial biogenesis and degradation. Meth significantly upregulated *NUDT8*, an enzyme that detoxifies oxidized guanine nucleosides that present due to increased ROS levels (Figure 3D) [60]. Meth also significantly decreased *MARCH5*, an E3 ubiquitin ligase that targets vital proteins that regulate mitochondrial fusion/fission for degradation [61]. This could decrease the overall mitochondrial content in the cell and increase ROS (Figure 3C).

We examined the impact of meth on overall metabolism. Meth altered the expression of NF-κB regulators and appeared to increase *NFKB1* protein (Figure 3D). There was a significant decrease in the negative regulator of NF-κB, *NLRP2*, and a significant decrease in the positive regulator, *IKBKB* (Figure 3D) [19]. Meth may increase NF-κB and signaling to enhance cell survival, which occurs in macrophages with higher baseline ROS [62]. There were also trends with meth treatment toward decreased mTOR and Akt1 (Figure 3D). Decreased activity of mTOR/Akt1 also classically induces autophagy [44].

### 3.4. Meth Increases Protein Pathways Involved in Macrophage Functions and Neurodegenerative Diseases

Given the variability in raw abundance of each protein read across the primary human samples, we also analyzed differentially expressed proteins with *p* < 0.1, in addition to *p* < 0.05. By this method, there were 92 significantly upregulated proteins and 92 downregulates ones (Appendix A). Using these 184 proteins, we performed a KEGG pathway analysis and found significant changes in metabolism, oxidative phosphorylation, and phagocytosis. This confirmed that meth alters these already dysregulated functions in HIV-infected macrophages to potentially increase neuropathogenesis. Interestingly, the processes involved in several neurodegenerative diseases were also significantly enriched with meth, including Parkinson’s, Alzheimer’s, Huntington’s, and ALS (Figure 4A). Parkinson’s and Alzheimer’s pathways were enriched (FDR 0.0056 and 0.0405, respectively) even when analyzing proteins with *p* < 0.05. By STRING database interactome analysis, meth affects the regulation of viral processes, immune functions, and purine metabolism (Figure 4B). Viral processes and host–virus interactions were statistically enriched when analyzing *p* < 0.05 proteins, FDR 0.0337 and 0.0369, respectively. This map also highlights that neurodegeneration-related changes in the proteome may relate to changes in both oxidative phosphorylation and calcium signaling.

### 3.5. Meth Induces Autophagy, Inhibits Its Completion, and May Reduce p62 Degradation

Autophagic activity is a significant regulator of phagocytosis and oxidative metabolism. To measure the changes in autophagy, MDM were cultured from PBMC on 60 mm dishes, infected with HIV, and left untreated (Untx) or treated with meth and/or ART for 24 h. For the Western blotting experiments, 20 mM of NH4Cl + 200 µM of leupeptin (N + L) were added in the last 4 h of treatment to inhibit autophagic degradation or LC3II and p62 to measure flux. Lysates or RNA were isolated followed by Western blotting for LC3 (Figure 5B) and p62 (Figure 5F) and RT-qPCR analysis of *SQSTM1* expression (Figure 5J). Meth alone significantly increased LC3II levels only in the absence of lysosomal inhibitors (Figure 5C). This led to an overall trend toward decreased LC3II flux and no change in net flux (Figure 5D,E). There also appeared to be increased LC3II in the meth + ART treated samples, although this was not quite statistically significant. ART alone had no significant impact (Figure 5C,E). Meth appears to induce autophagy, but that induction does not increase autophagosome (APG) maturation. This causes LC3II, and potentially the associated APG cargo, to accumulate undegraded, which may further impair cell homeostasis.

One type of cargo degraded by autophagy includes polyubiquitinated proteins, most often linked to ubiquitin at K63 [63]. These proteins are docked to forming APG through the interaction of the cargo receptor, p62, with LC3II. Similar to LC3II, the autophagic flux of p62 can also be measured by Western blotting [29]. However, this pool of APG is small in comparison to total APG content, and only some donor’s cells have significant p62 flux at baseline. A total of 6 out of 10 LC3II experiments had measurable p62 flux in which significant accumulation was found in response to N + L (Figure 5G). Although meth and/or ART did not significantly change the baseline p62 levels, meth with or without ART did significantly reduce p62 flux and net flux (Figure 5H,I). This may reflect an overall decrease in the degradation of aggregated proteins due to impaired APG maturation. Transcription is also a significant mode of regulating p62 expression; thus, we hypothesized that changes in gene expression may explain why p62 does not accumulate significantly with meth treatment [31]. No statistically significant change in transcription was found after 6 h or 24 h of treatment, but there was a trend toward an increase at 24 h (Figure 5J). These changes could influence the protein levels of p62, and thus, the autophagic flux of polyubiquitinated cargo.

Using mass spectrometry data, we determined whether meth alters the expression of the proteins that regulate autophagy at the level of activation, suppression, and effector function. We also categorized these proteins as either involved in autophagy directly or involved in lysosomal function, with significance cutoffs of *p* < 0.1 and *p* < 0.05 (Appendix A). Meth significantly decreased *DAPK1*, a kinase that promotes autophagy by phosphorylating Beclin-1 to dissociate it from BCL-2 (Appendix A) [64]. Beclin-1, part of the class III PI3K complex essential for autophagy initiation and maturation, was not altered significantly, but there was a trend toward an increase (Appendix A) [65]. More antagonism by lower DAPK1 may be compensatory to higher Beclin-1. Conversely, meth significantly decreased Vps34 (*PIK3C3*), another key component of the PI3K complex with Beclin-1 [65]. While not significantly increased, there was a trend toward more *ATG5*, which is necessary for autophagy initiation (Appendix A) [66]. There was also a trend toward increased levels of *MAP1LC3B* that agrees with our Western blot data (Figure 5, Appendix A). However, correlating protein levels of LC3 with autophagic flux requires the specific analysis of the lipidated isoform, LC3II, as mentioned previously [30]. Unlike Western blotting, mass spectrometry cannot distinguish between LC3I and LC3II isoforms, which makes it less reliable for monitoring levels of autophagy.

Lysosomal function is vital for proper APG maturation and degradation of cargo to be recycled as building blocks for de novo organelle synthesis [67]. We analyzed whether meth affects the levels of several lysosomal proteins including enzymes, membrane proteins, acidifying proton pumps, and factors involved in lysosomal biogenesis. Meth statistically increased proton pumps, *ATP6V1C1* and *ATP6V1E1*, but this increase was not above our 1.2-fold change cutoff (Appendix A). This slight increase could mediate the effective autophagic degradation of select cargo, including aggregated proteins, lipids, or even organelles, such as mitochondria.

### 3.6. Meth May Induce Mitophagy to Decrease Total Mitochondrial Content

Mitophagy is a form of selective autophagy induced in response to increased ROS. There are several mechanisms by which depolarized, dysfunctional mitochondria can be degraded by autophagy, with the PINK1/Parkin pathway being the most studied [68]. We hypothesized that the degradation of mitochondria by autophagy, mitophagic flux, would be downregulated similarly to p62-mediated autophagy. By immunofluorescence and confocal microscopy Z-series studies of LC3 and TOM20, a mitochondrial outer membrane protein, we measured how meth impacts total autophagy and mitophagy in HIV-infected MDM (Figure 6A).

Ammonium chloride + L were added in the last 4 h of treatment to calculate flux. Different from LC3II Western blotting, LC3 puncta indicate the number of APG present rather than reflecting the total cellular content of LC3 [29]. By counting LC3 puncta in a Z-series per cell, there were no significant changes with or without N + L in response to meth (Figure 6B). This suggests that some APG may be maturing properly after meth induces autophagy, while other APG subgroups containing specific cargo may not mature appropriately. We observed a trend for decreased LC3 puncta positive for mitochondria, yellow puncta on overlay, representing mitophagosomes (Figure 6D). The trend did not persist with N + L, resulting in increased mitophagic flux (Figure 6E). This suggests that meth may selectively upregulate the autophagic degradation of mitochondria due to either direct or indirect impacts on mitochondrial function. Our quantification of the total TOM20 volume per cell showed that meth decreases the mitochondrial content, which is consistent with higher levels of mitophagy (Figure 6C) [69].

### 3.7. Drug Inhibited Lysosomal Degradation Increases ROS in Meth-Treated Macrophages

ROS induce total and selective autophagic processes [70]. We examined whether increasing or decreasing total autophagic activity during meth exposure further alters ROS levels. We treated HIV-infected MDM with meth for 24 h. In the last 4 h of treatment, for one set of cells we added leupeptin, which prevents lysosomal degradation of autophagic substrates, as shown in Figure 4. To another set of cells, in the last 3 h of treatment, we added 500 nM of Torin 2, an mTOR inhibitor that induces autophagy [71]. This timing and concentration of Torin 2 induced autophagy as shown by Western blotting, demonstrating higher LC3II flux and decreased p62 (Figure 7C). Torin 2 alone significantly increased ROS, although minimally, while leupeptin alone had no effect. However, meth + leupeptin significantly increased ROS relative to meth alone. Meth + Torin 2 had the same effect as meth alone (Figure 7A,D). However, these effects did not persist with oxidative stress induced by CCCP (Figure 7B,E). These data do not support our initial hypothesis that increasing autophagic activity could ameliorate meth-induced ROS. They also suggest that meth increases ROS through mechanisms independent of autophagic activity and underscore that inhibiting autophagy may enhance oxidative stress and resultant cell toxicity during meth exposure.

## 4. Discussion

Substance use disorder can worsen NCI [9]. The mechanisms by which HIV-NCI occurs in the era of ART are not fully characterized. Whether regular methamphetamine (meth) use impacts these mechanisms remains important to determine, given the rise in prevalence and incidence of meth use nationally [72]. Mature monocytes are preferentially infected with HIV and primed to cross the blood–brain barrier to reseed the CNS with HIV, even in PWH stably suppressed on ART [73,74]. These monocytes can become CNS-resident perivascular MDM that perpetuate infection and neuroinflammation to injure parenchymal cells [45].

In the CNS of people with HIV-NCI, MDM are heterogeneously infected, with some cells having HIV integrated into their genomes, termed HIV-positive cells, while others do not have integrated HIV but are still exposed to viral proteins, ROS, and inflammatory mediators. These are termed exposed cells [14,73]. In our functional and proteomic assays, we tested the impact of meth on HIV-infected MDM that contain HIV-positive cells and exposed cells. Based on our results and those of prior proteomic studies, we propose that meth impacts some macrophage functions regardless of HIV infection, while other functions meth alters require HIV infection.

We tested whether meth and/or ART change phagocytosis of a neurotoxic protein, aggregated amyloid-β [18]. Meth significantly decreased amyloid-β accumulation in HIV-infected MDM, and ART appeared to minimize the impact of meth on phagocytosis [16]. The levels of infection were similar in ART-treated cells and meth + ART-treated cells. ART may increase phagocytosis independently of HIV infection and in the context of phagocytic suppression by meth. These data indicate that taking a full ART regimen as prescribed could ameliorate the negative impact of meth on phagocytosis of amyloid-β. Meth may also accelerate the intracellular degradation of amyloid-β once internalized, but our proteomic data did not indicate significant increases in lysosomal proteases or proteins involved in phagosome maturation (Appendix A).

Phagocytosis requires several classes of molecular factors that recognize substrates, internalize them, and traffic phagosomes [50]. Our mass spectrometry results indicate that meth may impact all these components. The most significantly changed proteins **are** involved in actin cytoskeleton disorganization [42]. Other groups that performed proteomic analyses of primary human MDM demonstrated that HIV infection can either dampen phagocytosis to promote viral replication or activate infected cells to phagocytize microbes. This occurs by regulating the factors that reorganize the actin cytoskeleton, primarily profilin 1 and cofilin 1 [75,76]. Both proteins were highly expressed by HIV-infected MDM in our study**,** but were unchanged with meth (Appendix A). Another study demonstrated that meth inhibits phagocytosis in uninfected dendritic cells [35]. Together, these data suggest that meth reduces phagocytosis independently of HIV infection. Inhibited phagocytosis could be caused by decreased PI(3)P as a result of decreased class III PI3K activity (gene products of *PIK3C3* and *PIK3R1*) and/or decreased recognition of substrates by scavenger receptors, *MSR1* and *SCARB1* (Figure 2C,D). However, CD16, which recognizes the opsonized substrates to be engulfed, was increased with meth [77,78]. This could increase substrate recognition and impact downstream signaling.

We also measured reactive oxygen species (ROS) [79]. Meth treatment of HIV-infected macrophages significantly increased ROS, even when higher levels of baseline stress were induced by a mitochondrial uncoupler, CCCP. Our ROS experiments in uninfected MDM exposed to meth and/or ART yielded the same results (*n* = 3, Appendix A). This demonstrates that meth causes redox changes, regardless of HIV infection. In our studies, we measured total ROS in the cell, which are reflective of the pathways including cell respiration, phagocyte NADPH oxidase activity, and downstream neutralization of ROS [19]. By proteomics, the levels of proteins involved in the respiratory burst from NADPH oxidase were unchanged, as well as any isoform of superoxide dismutase (Appendix A). Thus, we hypothesized that meth increases mitochondrial ROS, although this was not specifically measured.

Due to its affinity for protons, we predicted that meth disrupts the proton gradient across the inner mitochondrial membrane to increase ROS [36]. We analyzed mitochondrial membrane potential by a JC-1 assay. Meth did not alter the monomeric or aggregate mitochondrial signal (Appendix A). These data indicate that meth may increase ROS through more indirect mechanisms related to changes in signaling, and perhaps, gene expression. However, we did not specifically measure mitochondrial ROS generated from altered ETC function. Future studies will address this. By mass spectrometry, meth changed ETC components and other factors that contribute to altered oxidative phosphorylation and mitochondrial degradation. There was also an increased expression of the enzymes involved in oxidative metabolism, as well as crucial signaling factors that regulate oxidative status, including higher NF-κB and its regulators, and trends toward lower mTOR and Akt1 [19]. Proteomic studies of the secretome of HIV-infected primary human MDM demonstrated significantly increased superoxide dismutase. This was confirmed by Western blotting [76]. Thus, active neutralization of ROS in infected MDM in response to meth may also contribute to oxidative stress.

ROS cause rapid processing by NLRP3 inflammasomes and secretion of IL-1β, which is increased in the CNS of PWH with cognitive impairment [6,80]. Two types of factors are needed to activate these inflammasomes to convert cytokine pro-forms into active cytokines. The first is a pathogen-associated molecular pattern, such as bacterial LPS, and the second is a danger-associated molecule, such as ATP [81]. We predicted that more long-term treatment with meth could increase the activation of these inflammasomes in response to LPS and ATP and increase IL-1β secretion. LPS + ATP effectively activated inflammasomes (Appendix A). However, daily treatment with meth and with or without ART for 7 days did not significantly change IL-1β (Appendix A). Meth may alter the levels of other inflammatory cytokines implicated in HIV pathogenesis. By proteomics, meth significantly increased CXCL16, a ligand for CXCR6, a coreceptor for the entry of the simian immunodeficiency virus (SIV) into PBMC, mostly T cells (Appendix A) [82]. Higher levels of HIV infection in macrophages also increase the elaboration of CXCL16 [82,83]. A proteomic analysis of uninfected primary human MDM also demonstrated that meth increases the secretion of CXCL16 [84]. This indicates that meth increases this cytokine independently of HIV infection. Future studies will address how increased CXCL16 contributes to HIV pathogenesis in MDM exposed to meth.

Changes in ROS and autophagic activity significantly contribute to neurodegeneration [67]. Autophagy negatively regulates the inflammasome-mediated release of IL-1β [22,85]. We measured how meth changes total autophagy by LC3, selective degradation of poly-ubiquitinated proteins by p62, and mitophagy. ART alone had no significant effect on any parameters related to LC3II or p62 according to Western blotting. Therefore, we did not examine its impact on mitophagy. Meth induced autophagy in HIV-infected MDM, but APG maturation did not increase similarly, causing significant accumulation of LC3II. Increased induction of autophagy by meth is consistent across cell culture and animal models, which can contribute to autophagic cell death in certain cell types. Other studies showed that toxicity from meth is reduced when autophagy is attenuated pharmacologically [86,87,88]. The dual impact of meth on autophagy induction and completion in our system is similar to how neuronal autophagy is altered in Alzheimer’s [89]. We performed the same Western blotting experiments in uninfected MDM treated with meth and found no significant changes in autophagy (*n* = 3, Appendix A). This suggests that meth impacts autophagy in an infection-dependent manner.

One type of selective autophagic cargo includes poly-ubiquitinated proteins, which are docked by the receptor, p62 [63]. We found a significant decrease in the rate and amount of p62 being degraded with meth, even with concomitant ART. Our proteomic analyses also did not suggest that p62 was increased with meth (Appendix A). However, we performed the same Western blotting experiments in uninfected MDM and found an increase in p62 steady state levels and decreased flux, as we found in infected cells (Appendix A). Our previous studies demonstrated that HIV infection increases the baseline levels of p62 and decreases flux [33]. This p62 may not accumulate as much in infected MDM in response to meth because steady state levels are high at baseline.

The levels of p62 can be regulated by oxidative stress. Excess cytosolic p62 binds Keap1, which sequesters the transcription factor, Nrf2, and causes translocation to the nucleus to activate the transcription of antioxidant genes [90]. The *SQSTM1* gene is itself a target to increase the responses related to autophagy and apoptosis [85]. We found a trend toward increased *SQSTM1* transcription with 24 h of meth treatment but not at an earlier time point, 6 h. *SQSTM1* transcription may be dampened until around 24 h. Our ROS studies show that at this later time point of meth treatment, there are more ROS, which could activate Nrf2. Analyses of autophagy at time points after 24 h could further characterize how meth impacts these processes to contribute to neurodegeneration.

Our pathway analyses of differentially expressed proteins suggested several metabolic and disease-related processes that meth impacts in HIV-infected macrophages. Viral processes and host-virus interactions were also significantly enriched. Most of the involved proteins mediate the transcriptional and translational regulation of RNA virus expression and restriction (Figure 4B). This suggests that gene expression-related mechanisms may explain the existing in vitro data demonstrating that meth pre-treatment can enhance HIV infection [48,91]. Our data similarly highlight an interaction between meth and HIV in infected macrophage reservoirs that could mediate clinical findings of higher inflammatory markers in meth using PWH [13]. Proteins associated with neurodegenerative diseases, such as Alzheimer’s and Parkinson’s, were significantly enriched. This underscores macrophages as key mediators of CNS damage in multiple disease processes. These specific proteins are involved in oxidative metabolism, proteasomal activity, and signaling through calcium and PI3K (Figure 4B). Therefore, therapies for known neurodegenerative conditions that target these processes may also improve outcomes in PWH using meth with cognitive impairment.

Gene expression analyses of human macrophages treated with meth could determine the transcriptional pathways that meth alters to mediate differential protein expression. In vivo studies of gene expression in SIV-infected macaque brains exposed to meth found an increased proportion of infected macrophages, which is consistent with our mass spectrometry data and prior in vitro studies [92,93]. These analyses demonstrated the differential expression of genes in the complement pathway and electron transport chain function [93]. Both immune and metabolic processes, coined “immunometabolism,” are likely to be key in the changes we measured in our functional assays. Preferences for aerobic metabolism in M2 macrophages or anaerobic metabolism in M1 have been characterized, although inflammatory activity and its relation to M1 and M2 is nuanced [94]. scRNA-seq analyses found that glycolytic and pentose phosphate pathways were enriched in SIV-infected brain macrophages exposed to meth [93]. We also found significant increases in four MHCII proteins, as well as higher ROS, which correspond with the more inflammatory M1 subtype (Appendix A, Figure 3). Cytokines were mostly not detected by mass spectrometry. There were also no changes in major M2 surface markers, CD206 and CD163, or the M1 marker, CD32 [94,95]. The studies of macrophage cell lines treated with meth similarly demonstrated a more M1-like subtype characterized by increased nitric oxide and IL-12 [96]. Future metabolomic studies involving Seahorse, LC/MS/MS, and stable isotope-labeled carbon flux analyses may determine how meth alters specific metabolic activities to establish a more inflammatory state.

Targeting the signaling changes caused by meth may also provide therapeutic benefits, as suggested by the enrichment of Parkinson’s pathway-related proteins. Meth appears to increase NF-κB activation, but decreases levels of class III PI3K, and perhaps, mTOR and Akt1. Dampened Akt1 is associated with M1 macrophages [97]. Additional studies will determine the mechanisms by which these survival and metabolic factors are changed by meth. These mechanisms may relate to the upregulation of calcium signaling factors, such as calmodulins and calmodulin-dependent kinases (Figure 2D). Altered intracellular calcium is a major output of signaling through the sigma-1 receptor, which is present on the ER [98]. Several studies indicate that meth activates these receptors in microglia [99,100,101]. According to mass spectrometry, meth increased sigma-1 receptor expression in all four experiments, with a mean fold change of 1.172 relative to the control, *p* = 0.0597 (Appendix A). The ranked expression of this receptor in HIV-infected macrophages at baseline was high and consistent as well, ranging from 1527 to 1745 out of 3655 total reads. Future studies will examine whether altering sigma-1 receptor activation ameliorates the dysregulation of macrophage functions [102].

Our studies demonstrate that meth dysregulates key functions of HIV-infected macrophages in the CNS, including phagocytosis, total ROS, and autophagy. Based on our mass spectrometry data and others, some of these effects are mediated by meth alone, while others are due to the interaction between meth and HIV infection. These data suggest interventional strategies to reduce neurocognitive deficits in people with HIV-NCI who use meth that target the potential pathways of calcium signaling, PI3K activity, NF-κB, and/or sigma-1 receptor activity.

## Figures and Tables

**Figure 1 biomedicines-10-01257-f001:**
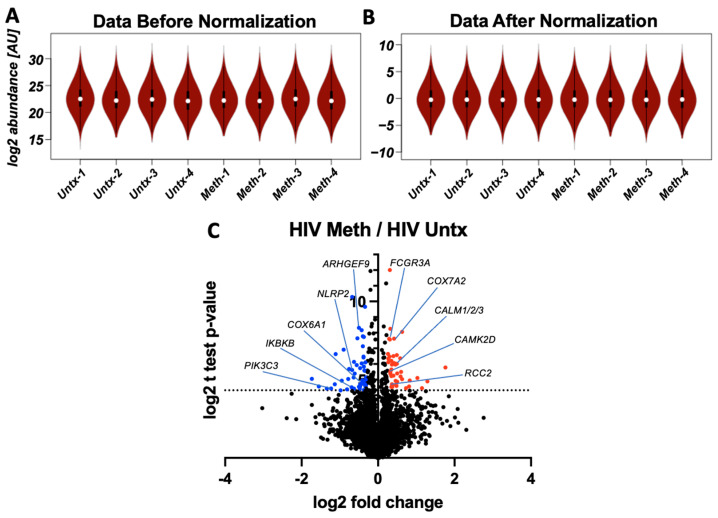
Mass spectrometry proteomics analysis performed on primary human HIV-infected MDM treated with and without methamphetamine (meth, 50 µM) for 24 h from four individuals. (**A**) Distribution of protein abundances across samples after log2 transformation and (**B**) after normalizing the dataset by the center of the distribution, i.e., average. The similarity between the two graphs indicates that the samples were injected at very similar amounts, thus, not requiring advanced normalization steps. (**C**) Volcano plot displaying the log2 fold change (*x*-axis) and the log paired *t*-test *p*-value (*y*-axis) with indicated upregulated proteins (in red, fold change of >1.2, *p* < 0.05) and downregulated proteins (in blue, fold change of <0.8, *p* < 0.05).

**Figure 2 biomedicines-10-01257-f002:**
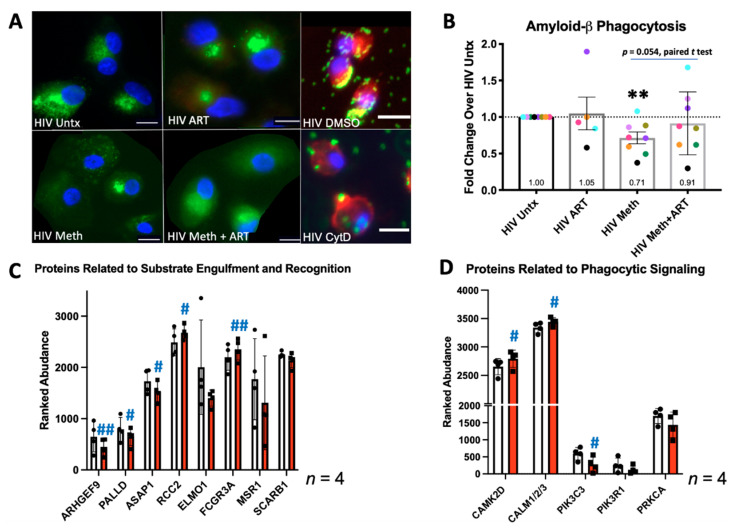
Phagocytosis of 1–42 amyloid-β by HIV-infected MDM. Macrophages were infected with HIV for 3–4 days, treated with methamphetamine (meth, 50 µM) and/or ART (16 nM of tenofovir, 441 nM of emtricitabine and 43 nM of dolutegravir) for 24 h, and in the last two hours, fluorescent 1–42 β-amyloid (green) was added prior to fixation, mounting, and visualization. Control cells were pre-incubated with 20 µM of cytochalasin D (CytD) for 20 min and was present throughout phagocytosis. (**A**) Representative images of untreated (HIV Untx), HIV ART, HIV, Meth, and HIV Meth + ART with HIV DMSO and HIV CytD as controls. HIV DMSO and HIV CytD cells were counterstained with Alexa 594 conjugated phalloidin (red). (**B**) Fold change in median accumulation of amyloid-β per cell during phagocytosis, with HIV Untx set to 1.0 in each experiment. (**C**,**D**) Differentially expressed proteins detected by mass spectrometry related to phagocytic substrate recognition, engulfment, and signaling, HIV Untx is in white bars and HIV Meth is in red bars. *n* = 4–8 with 50–75 cells counted per treatment in each microscopy experiment ** *p* < 0.01 one-sample *t* test, ^#^ *p* < 0.05 ^##^ *p* < 0.01 paired *t* test. Scale bar is 15 µm. Error bars represent SEM for (**B**) and SD for (**C**,**D**).

**Figure 3 biomedicines-10-01257-f003:**
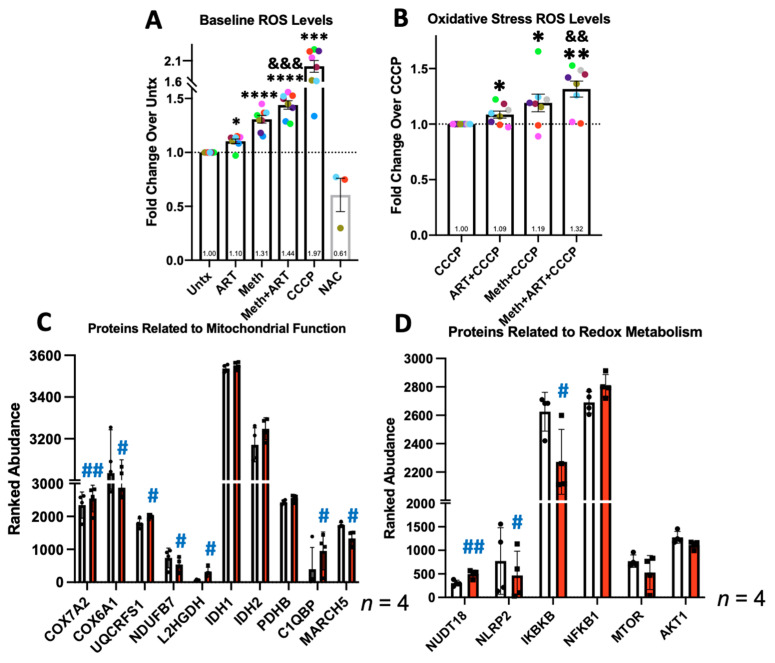
Total reactive oxygen species (ROS) in HIV-infected MDM were detected by a fluorometric plate reader assay. CM-H2DCFDA dye in culture media was added for 1 h followed by incubation in PBS, 30 µM of CCCP, or 5 mM of N-acetylcysteine (NAC) for 1 h, followed by detection of fluorescence by fluorometry. (**A**) ROS levels at baseline without stimulation with untreated cells (Untx) set to 1.0 in each experiment. (**B**) ROS levels in cells stimulated with CCCP for 1 h with CCCP alone set to 1.0 in each experiment. (**C**,**D**) Differentially expressed proteins detected by mass spectrometry related to mitochondrial function and cellular redox metabolism, HIV Untx in white bars and HIV Meth in red bars. *n* = 4–8; * *p* < 0.05, ** *p* < 0.01, *** *p* < 0.001, **** *p* < 0.0001, one-sample *t* test or Wilcoxon signed rank test. ^&&^ *p* < 0.01, ^&&&^ *p* < 0.001 Meth + ART compared to ART alone by one-way ANOVA. ^#^ *p* < 0.05, ^##^ *p* < 0.01 paired *t* test. Error bars represent SEM for ROS data and SD for proteomics data.

**Figure 4 biomedicines-10-01257-f004:**
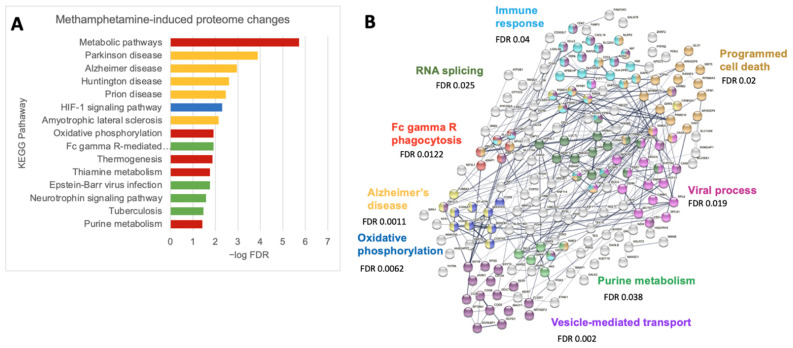
Pathway analyses and interactomes by proteomics in meth-treated HIV-infected macrophages. (**A**) Significantly enriched proteins with fold change of either <0.8 or >0.2 and *p* < 0.1 were analyzed for pathway analysis using KEGG software. (**B**) Significantly enriched proteins with fold change of either <0.8 or >0.2 and *p* < 0.1 were analyzed for protein–protein interactions using the STRING database and highlighted according to matching processes.

**Figure 5 biomedicines-10-01257-f005:**
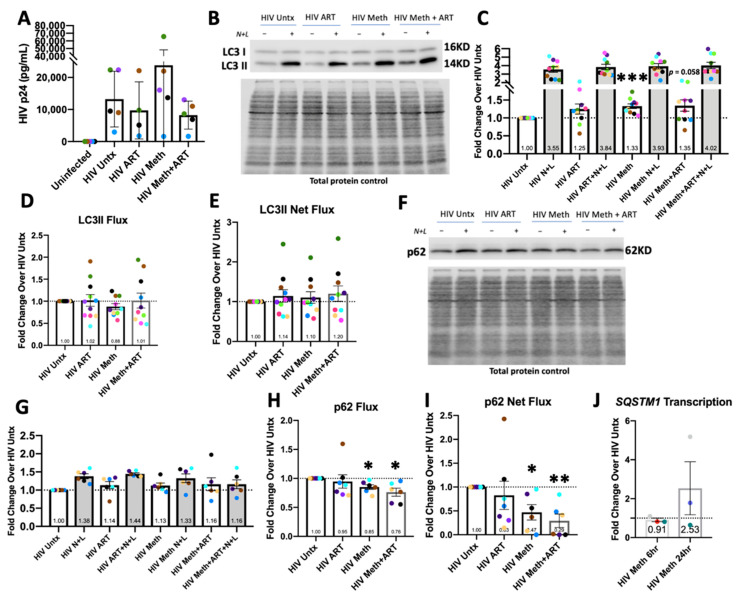
Normalized LC3II and p62 levels, LC3II or p62 flux, LC3II or p62 net flux, and *SQSTM1* transcription in HIV-infected macrophages. PBMC were cultured into MDM, infected with HIV-ADA as described, and left untreated (HIV Untx) or treated with meth and/or ART for 24 h. NH4Cl and leupeptin (N + L) were added directly to indicated cultures in the last 4 h. (**A**) HIV p24 values in pg/mL from supernatants in five experiments, including the untreated control (HIV Untx). (**B**) Representative Western blot for LC3I and LC3II. (**C**) LC3II levels normalized to total protein content and then to HIV Untx control set to 1.0 in each experiment. (**D**) LC3II flux calculated using LC3II values with HIV Untx set to 1.0. (**E**) LC3II net flux calculated using LC3II values with HIV Untx set to 1.0. (**F**) Representative Western blot for p62. (**G**) p62 levels normalized to HIV Untx set to 1.0 in each experiment. (**H**) p62 flux calculated using p62 values with HIV Untx set to 1.0. (**I**) p62 net flux calculated using p62 values with HIV Untx set to 1.0. (**J**) RT-qPCR for *SQSTM1* in HIV-infected MDM untreated or treated with meth for 6 h or 24 h. *n* = 3–10; * *p* < 0.05 ** *p* < 0.01 *** *p* < 0.001, one-sample *t* test. Error bars for (**A**) represent SD, all others represent SEM.

**Figure 6 biomedicines-10-01257-f006:**
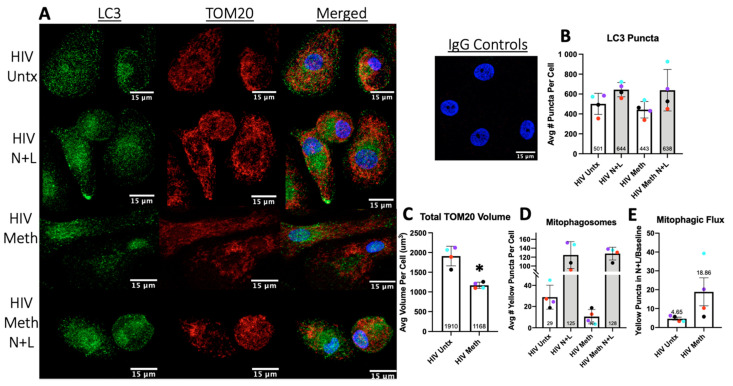
Confocal immunofluorescence studies of autophagosomes (LC3) and mitochondria (TOM20) in HIV-infected macrophages untreated (HIV Untx) or treated with meth for 24 h and N + L added in the last 4 h of treatment for indicated conditions. (**A**) Representative images of LC3 puncta (green), mitochondria (TOM20, red), and nuclei (DAPI, blue) and corresponding rabbit and mouse IgG controls. (**B**) Number of LC3 puncta per cell in a Z-series was quantified in Volocity according to the Section 2. (**C**) Total amount of TOM20 staining was quantified per cell. (**D**) Number of LC3 puncta positive for TOM20, mitophagosomes (yellow puncta), per cell were quantified in a Z-series. (**E**) Number of mitophagosomes in (**D**) was used to quantify mitophagic flux with and without meth treatment. *n* = 4 with 50–75 cells analyzed per condition per experiment; * *p* < 0.05, paired *t* test. Scale bar is 15 µm. Error bars represent SEM.

**Figure 7 biomedicines-10-01257-f007:**
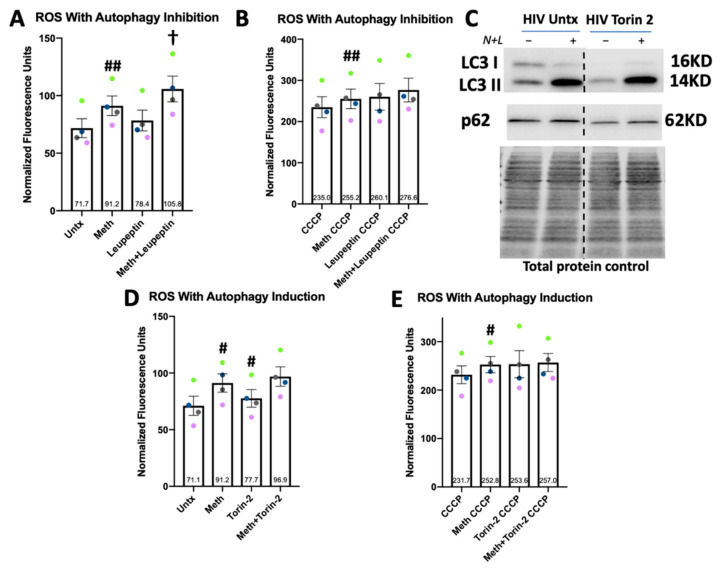
Meth-mediated upregulation of ROS is increased when lysosomal degradation is inhibited but not when autophagy is induced. (**A**) A total of 200 µM of leupeptin, which inhibits autophagic flux, was added in the last 4 h of 24 h treatment, and ROS were detected by fluorometry as described. (**B**) Same experiments as in (**A**) with CCCP alone as the untreated control. (**C**) Western blotting for LC3 and p62 in HIV-infected MDM treated with 500 nM of Torin 2, an mTOR inhibitor, for 3 h, demonstrating increased LC3II flux and decreased p62 levels. (**D**) A total of 500 nM of Torin 2, which induces autophagy, was added in the last 3 h of 24 h treatment, and ROS were detected by fluorometry as described. (**E**) Same experiments as in (**D**) with CCCP alone as the untreated control. *n* = 4, ^#^ *p* < 0.05, ^##^ *p* < 0.01 meth or Torin 2 alone compared to untreated by one-way ANOVA; ^†^ *p* < 0.05 meth compared to meth + leupeptin by one-way ANOVA. Error bars represent SEM.

## Data Availability

Not applicable.

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
