# Peer review of "Methamphetamine Dysregulates Macrophage Functions and Autophagy to Mediate HIV Neuropathogenesis"

_biomedicines, 2022, doi:10.3390/biomedicines10061257_

Round 1

Reviewer 1 Report

This is a very well conducted study. However, I feel that introduction and discussion section are too heavy and should be shortened. Furthermore the discussion ends abruptly; it requires a few closing and concluding sentences.

Reviewer 2 Report

The study is well-conceived and designed, well written and described; methods are properly chosen and rigorous. However, I have a main concern regarding this study, related to the role of HIV in the proteomic and functional changes described. The authors compared infected macrophages treated with meth with HIV-infected, untreated cells. However, there are no uninfected, control cells that can help to understand if the proteome effects attributed to Meth are present only in an HIV background, or if they are independent of HIV. I suggest adding an HIV- control or, if this is not possible, comparing HIV+ data with data reported in the literature and/or present in a public proteomic database.  This point should have at least been discussed.

Concerning methods:

  1. It is not clear how the authors differentiate monocytes into macrophages.
  2. Since the authors have focused their attention on mitochondria, and they found significant changes in the expression of OXPHOS related proteins, I think it could be more informative to distinguish ROS produced by mitochondria (e.g. superoxide anion, by using MitoSOX) from total ROS (which can include ROS produced by phagocyte oxidase). If it is not possible to perform this experiment, I suggest adding this point in the discussion, as a limitation of the study.

Concerning results:

Can the observations made by the authors be dependent on the phenotype/differentiation status of macrophages? Have the authors checked the typical markers of M1/M2  macrophages (being aware of the limitation of M1/M2 dichotomy)?
